

# Development, characterization and first deployment of an improved online reactive oxygen species analyzer

**Jun Zhou[1], Emily A. Bruns[1], Peter Zotter[2], Giulia Stefenelli[1], André S. H. Prévôt[1], Urs Baltensperger[1],**

**Imad El-Haddad[1] & Josef Dommen[1]**

[1] Paul Scherrer Institute, Laboratory of Atmospheric Chemistry, 5232 Villigen PSI, Switzerland

[2] Lucerne University of Applied Sciences and Arts, School of Engineering and Architecture, CC Thermal Energy Systems & Technology, Bioenergy Research, 6048 Horw, Switzerland

*Correspondence to*: Josef Dommen (josef.dommen@psi.ch)

**Abstract.** Inhalation of atmospheric particles is linked to human diseases. Reactive oxygen species (ROS) present in these atmospheric aerosols may play an important role. However, the ROS content in aerosols and their formation pathways are still largely unknown. Here, we have developed an online and offline ROS analyzer using a 2',7'-dichlorofluorescin (DCFH) based assay. The sensitivity of the ROS analyzer was characterized using a suite of model organic compounds. The instrument detection limit determined as three times the noise is 1.3 nmol $L^{-1}$ for offline analysis and 2 nmol $m^{-3}$ of sampled air when the instrument is operated online at a fluorescence response time of approximately 8 min, while the offline method detection limit is 9 nmol $L^{-1}$ to 13 nmol $L^{-1}$. Potential interferences from gas phase $O_3$ and $NO_x$, matrix effects of particulate $SO_4^{2-}$ and $NO_3^-$ were tested, but not observed. $Fe^{3+}$ had no influence on the ROS signal while soluble $Fe^{2+}$ reduced it if present at high concentrations in the extracts. Both online and offline methods were applied to identify the ROS content of different aerosol types, i.e., ambient aerosols as well as fresh and aged aerosols from wood combustion emissions. The stability of the ROS was assessed by comparing the ROS concentration measured by the same instrumentation online in-situ with offline measurements. We also analyzed the evolution of ROS in specific samples by conducting the analysis after storage times of up to four months. The ROS were observed to decay with increasing storage duration. From their decay behavior, ROS in secondary organic aerosol (SOA) can be separated into short- and long-lived fractions, with an average half-life of ~1.7 h and ~432 d, respectively. All these measurements showed consistently that, on average $60 \pm 20$ % of the ROS were very reactive and disappeared during the filter storage time. This demonstrates the importance of a fast online measurement of ROS.

## 1 Introduction





Aerosol particles have negative effects on human health (Pope and Dockery, 2006), with an estimated 3 % of cardiopulmonary and 5 % of lung cancer deaths attributable to particulate matter (PM) globally (WHO, 2013). One of the important pathways leading to deleterious impacts on health is believed to be induced oxidative stress by the generation of reactive oxygen species, through the interaction of particulate matter (PM) with the human lung (Donaldson et al., 2002). Reactive oxygen species (ROS) denote chemically reactive molecules

containing oxygen (e.g., radicals, oxygen ions and peroxides including OH radicals, $O_2^{\cdot-}$, $H_2O_2$, organic peroxides and transition metals) (Fuller et al., 2014; Sagai et al., 1993). As one of the main free radical sources generated in our body by various endogenous systems, ROS can adversely alter lipids, proteins, and DNA structures, potentially leading to aging and numerous human diseases (Devasagayam et al., 2004). ROS exist both in the gas-phase and in PM. ROS are found inside the human body either through generation by the inhaled PM in vivo (endogenous ROS), or by transportation into the lungs on respirable particles (exogenous ROS) (Zhao and Hopke, 2012). While gas

phase ROS are most likely removed in the upper mucus membranes through diffusion (Kao and Wang, 2002), ROS associated with fine particles can penetrate deeply into the lungs, causing oxidative stress and cell damage. Understanding the mechanisms by which ROS are formed, evolve and decay in the atmosphere is therefore of utmost importance for mitigating their influence on human health (Khurshid et al., 2014).

Currently, many acellular assays exist for the determination of ROS quantities in particles, including dithiothreitol (DTT) (Fang et al., 2015)

and 2,7-dichlorofluorescin (DCFH) (Fuller et al., 2014; King and Weber, 2013a). The DCFH assay is one of the most commonly used assays today. Accurate ROS quantification remains challenging because some ROS species are highly reactive and are likely at least partially degraded prior to measurement when using offline techniques, which typically have delays of hours, days or weeks. Therefore, online techniques (through direct sampling into the liquid phase and measurement within a few minutes) are necessary for reliable ROS quantification (Wragg et al., 2016).

In this work, we developed and characterized a highly sensitive ROS analyzer which can be used either online or offline. The removing efficiency of interfering oxidizing trace gases of $O_3$ and $NO_x$ was tested, and the matrix effects of particulate $SO_4^{2-}$ and $NO_3^-$, as well as transition metals were assessed. Results from the application of this online and offline methodology to laboratory measurements of wood combustion emissions and ambient measurements at an urban site in Bern (Switzerland) are presented. To assess the stability of ROS, online in-situ measurements were compared with offline measurements using the same instrumentation, and the evolution of ROS on specific

samples was evaluated by conducting the analysis after storage times of up to four months. The results are put into perspective of future ROS measurement strategies.





## 2 Methods

### 2.1 ROS analyzer

In our experiments, ROS were measured using a DCFH assay, which is commonly used for examining ROS generation at a cellular level but has also been used for determining the oxidation potential (OP) of PM as an acellular assay (Fuller et al., 2014; King and Weber, 2013a;

Perrone et al., 2016; Sauvain et al., 2013; Venkatachari et al., 2007; Venkatachari et al., 2005). In this assay, the presence of oxidizing species is assessed from the rapid oxidation of DCFH to the fluorescent compound dichlorofluorescein (DCF), in the presence of horseradish peroxidase (HRP). The chemical reaction mechanism is shown in Fig. S1.

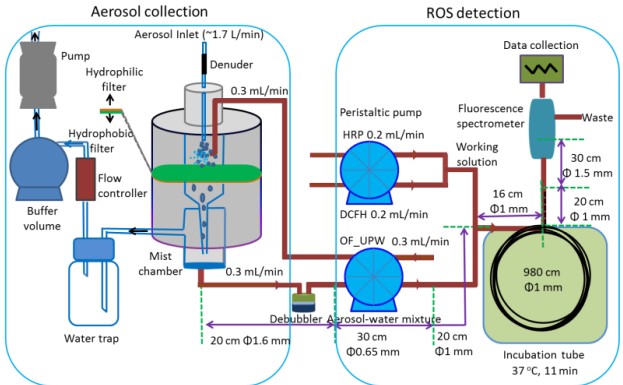

**Figure 1.** An overview of the online ROS analyzer, OF-UPW refers to oxygen free-ultra pure water. The same setup without the aerosol collector was used
for the offline analysis (shown in Fig. S2).

A schematic of the online aerosol ROS analyzer is shown in Fig. 1. The analyzer is composed of 3 compartments: the aerosol collector, the reaction and incubation region and the fluorescence analyzer. The same setup without the aerosol collector was used for offline analysis.

### 2.1.1 Aerosol collection

Particles were collected at a flow rate of ~1.7 L min$^{-1}$, using a particle into liquid extraction system (PILS, also called mist chamber/aerosol

collector; (Takeuchi et al., 2005)). Before the PILS, a honeycomb charcoal denuder of 10 cm length with 7 mm outer diameter (36 % open area; 450 μm channel width) was installed inside a stainless steel tube to remove $O_3$, $NO_x$, and organic vapors. The denuder was regenerated for at least 24 h at 250 $^{\circ}$C under a stream of 99.999 % $N_2$ before each experiment. By using at least two denuders, we were able to switch between them and to perform the experiments continuously.

The Plexiglas mist chamber had an approximate volume of 13.5 cm$^3$. It consisted of an air inlet, a nebulizing nozzle inlet port for pure water injection, a 2.5 cm diameter hydrophilic cellulose filter (Grade 497 circles, Schleicher & Schuell Rundfilter) supported by a 5.0 µm pore size hydrophobic membrane filter (Isopore membrane filters, TMTP, Millipore) to prevent the loss of the sample solution, an outlet to the vacuum pump and an exit for the water extracts (Fig. 1). Between the vacuum pump and the mist chamber, a flow controller protected by a

water trap was installed. To stabilize the air sampling flow, an additional gas buffer volume was introduced before the pump. The oxygen-free ultra-pure water (OF-UPW, 18.2 MΩ cm at 25 °C, total organic carbon (TOC) < 3 ppb), prepared by bubbling 99.999 % N$_2$ for ~ 20 min to reduce the dissolved oxygen, was used to collect the water soluble aerosol. The use of oxygen-free water reduced the instrument background by a factor of ~2 compared to normal ultra-pure water. The 1.7 L min$^{-1}$ air stream was mixed with the OF-UPW which was sprayed into the mist chamber with 0.3 mL min$^{-1}$ where the aerosol particles were incorporated into the water droplets. The liquid containing

the water soluble fraction of the aerosol, was collected at the bottom of the mist chamber for analysis.

In most studies using the DCFH-assay, aerosol samples were extracted in either in a DCFH-HRP (King and Weber, 2013b) or an HRP solution (Fuller et al., 2014). These approaches either shorten the usable lifespan of the working solution (WS) or induce additional contamination during the sample transport in the ROS analyzer. Therefore, we used only OF-UPW to extract the aerosol samples. The DCFH and HRP reagents were kept separate and only mixed together right before the aerosol aqueous extract was added.

**2.1.2 ROS detection**

The aerosol aqueous extract collected from the PILS was sampled by a peristaltic pump through a TRACE TRAP debubbler (TRACE Analytics GmbH, Germany), which effectively removed gas bubbles in the sample liquid without introducing a large dead liquid volume and signal broadening. At the same time, the two reagent solutions DCFH and HRP were drawn by another peristaltic pump and mixed to form the WS. The aerosol aqueous extract was then mixed with the WS and pumped through a reaction coil consisting of PEEK tubing (9.8

m length 1.6 mm OD, 1.0 mm ID, Kinesis GmbH) in an air ventilated temperature controlled housing held at 37 °C. The obtained solution was then analyzed using a spectrofluorimeter with excitation and emission wavelengths of 470 nm and 520 nm, respectively. All the transparent parts of the system were wrapped with aluminum foil to avoid the photooxidation of the DCFH.

**2.1.3 Offline analysis**

The instrument was also used for offline analysis of filters (Fig. S2). Filter punches were extracted in a vial with OF-UPW for 15 min at

30 °C. The vial was then vortexed (Vortex Genie 2, Bender& Holbein AG, Switzerland) for 1 min to ensure homogeneity and filtered



through a 0.45 μm nylon membrane syringe filter (Infochroma, Switzerland), prior to mixing with the WS and analysis in the fluorescence spectrometer.

Often, filters are extracted in an ultrasonic bath. However, recent studies suggest that sonication of pure water with dissolved air may create hydroxyl radicals due to the high temperature and pressure created by the collapse of bubbles formed by cavitation, which then form $H_2O_2$

or react with sample species (Mark et al., 1998; Miljevic et al., 2014). This was also demonstrated by Fuller et al. (2014), who showed the formation of 0.08 nmol m$^{-3}$ ROS by the sonication of pure water. These effects have also been confirmed in our laboratory, by analyzing filters collected at an urban site in Milan extracted with and without sonication (Perrone et al., 2016). Therefore sonication was not used for filter extraction during offline measurements.

**2.1.4 Working solution**

The stability of the WS is an important factor. Since HRP can catalyze the reaction of DCFH with dissolved oxygen in the phosphate buffer (Berglund et al., 2002; Huang et al., 2016; Rota et al., 1999a; Rota et al., 1999b), the phosphate buffer solution (PBS, 1 M; Sigma, USA) was degassed with 99.999 % $N_2$ for ~20 min. Furthermore, the two reagents DCFH and HRP were prepared separately as follows:

DCFH reagent: 2′,7′-dichlorofluorescin diacetate (DCFH-DA) (0.61 mL, Sigma-Aldrich, USA) stock solution (0.001 M) was mixed with NaOH (10 mL, 0.001M, Sigma-Aldrich, USA) for 30 min under dark conditions to initiate a deacetylation at room temperature. Then PBS

(25 mL) was added to set the solution pH at 7.2 and neutralize any remaining NaOH. This produces the fluorescent probe DCFH, referred to as WS (a) hereafter.

HRP reagent: Horseradish peroxidase (0.44 mg, HRP, type II, Sigma-Aldrich, USA) was dissolved in PBS (35.6 mL) to generate a stock solution of 2 units mL$^{-1}$, which is referred to as WS (b) afterwards.

WS (a) and WS (b) were then degassed for 20 min and only mixed together during the analysis at a 1:1 ratio. The final WS was 17.6 μM of

DCFH and 1 unit mL$^{-1}$ of HRP. This WS and the applied procedures provided the following advantages compared to previous analyzers using the same assay:

(1)  The pH of the WS was maintained constant at 7.2, which resulted in a stable background.

(2)  HRP and DCFH were prepared separately and mixed together only right before the combination with the sample solution. This reduced auto-oxidation and decreased the instrument background signal.





(3) Both working solutions were stored at ~ 4 °C and could be used for up to 1 week, while a mixed DCFH-HRP is not stable for more than

one day.

**2.1.5 Calibration**

The instrument was calibrated with known concentrations of $H_2O_2$ solutions. Standards were prepared from a concentrated solution of

hydrogen peroxide ($H_2O_2$; Sigma-Aldrich, solution, 3 wt. % in water). Calibration solutions of different concentrations were generated by

diluting different amounts of a stock solution with OF-UPW. The blank values were obtained by measuring OF-UPW alone.

For the online operation mode, $H_2O_2$ equivalent particulate ROS concentrations were determined as follows:

$$C\left(\frac{nmol}{m^3}\right) = \left(\frac{I-b}{a}\right)\left(\frac{V_i}{Q_c}\right)$$ Eq. (1)

*I:* fluorescence signal (volt)

*b:* Calibration intercept from the linear regression fit

*a:* Calibration slope from the linear regression fit (Volt nM$^{-1}$)

$V_i$: OF-UPW flow into the mist chamber (mL min$^{-1}$)

$Q_c$: air flow through the mist chamber (L min$^{-1}$, at ambient temperature and pressure)

For the offline operation mode, particulate ROS concentrations in air was determined as follows:

$$C\left(\frac{nmol}{m^3}\right) = \left(\frac{I-b}{a}\right)\left(\frac{V_i}{Q_c}\right)\left(\frac{A_{filter}}{A_{punch}}\right)$$ Eq. (2)

$V_i$: volume of OF-UPW for filter extraction (mL)

$Q_c$: total air flow through the filter (L, at ambient temperature and pressure)

$\frac{A_{filter}}{A_{punch}}$: Ratio of the area of the entire filter to the area of the filter punch.

The instrument background of the online operation mode was always higher than that of offline operation mode which may be due to the

uptake of oxygen in the mist chamber in the online system.



**2.2 Instrument testing**

In order to assess the performance of the ROS analyzer, several tests were performed, including the following:

1) The influence of the incubation time and the instrument detection limit, repeatability and reproducibility (Sect. 3.1.1 and 3.1.2).

2) Response of the DCFH assay to selected compounds with expected oxidative potential (Sect. 3.1.2).

3) Assessment of the interference from selected abundant gas phase and PM constituents (Sect. 3.2 and 3.3) on the ROS signals.

4) Verification of the instrument performance using genuine aerosol samples. Measurement of the ROS content in ambient aerosols was performed offline using filter samples collected in Milan (Italy), San Vittore (Switzerland) and Bern (Switzerland) and online using the developed ROS analyzer in Bern (Switzerland) (Sect. 3.1.3, 3.4.1 and 3.4.2). These samples include total suspended particles (TSP), $PM_{2.5}$ and $PM_{10}$ (particulate matter with a diameter smaller than 2.5 μm and 10 μm, respectively). Laboratory samples were also measured, including online and offline ROS measurements of fresh and aged aerosols from wood combustion emissions, using two different aging tools, a potential aerosol mass chamber (PAM) and a smog chamber (SC). Tests aimed at the verification of the instrument linearity, the assessment of matrix effects, the comparison of online and offline ROS measurements and the examination of the ROS degradation.

**3 Results**

**3.1 Instrument performance**

**3.1.1 Reaction time and detection limit**

The reaction time between the WS and the aerosol sample is an important parameter. Here, reaction times of 11 and 22 min were investigated by using different reaction tube lengths in the incubation region and followed by measuring the fluorescence intensity resulting from the reaction of $H_2O_2$ (Fig. 2a) and 2-hydroperoxy-2-(2-hydroperoxybutan-2-ylperoxy) butane (Fig. 2b) with the WS. The 22 min incubation time resulted in a 35% higher instrumental background signal than the 11 min incubation time. However, the same incremental increase in fluorescence intensity was found for the sample solutions of both $H_2O_2$ and the organic peroxide at the two reaction times, resulting in the same detection sensitivity. Here the detection sensitivity (V $nM^{-1}$) is defined as the ratio between the change in the output signal (in volt) to the corresponding change in the peroxide concentration (in nM). This suggest that the fluorescence response is unaffected by the reaction time in the investigated range, even for compounds protected by tert-butyl groups. Therefore, an incubation of 11 min seems to be sufficient to reduce all peroxides that can react with DCFH and consequently we used this incubation time for the further experiments.



The residence time (from the time of injection to the fluorescence measurement) of the sample liquid in the ROS instrument system and the fluorescence response time (rise time, from 10 % to 90 % of the signal) was approximately 19 min and 8 min for the 11 min reaction time, respectively.

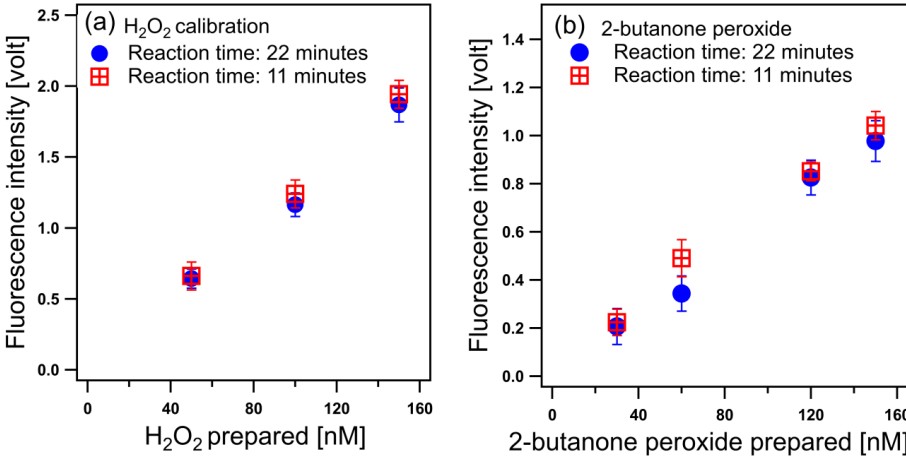

**Figure 2.** Fluorescence responses to (a) $H_2O_2$ and (b) 2-butanone peroxide under different reaction time. Error bars represent the propagation of the uncertainty ($\delta = \sqrt{{\delta_1}^2 + {\delta_2}^2}$, with $\delta_1$ representing the standard deviation of the instrument background signal of that experiment day, and $\delta_2$ the standard deviation of the sample signal.)

Under normal instrument operation condition, an instrument limit of detection (LOD) of 2 nmol m$^{-3}$ of sampled air was determined for the online methodology. This was obtained as three times the standard deviation when a particle filter was placed in the sampling line upstream of the analyzer (Long and Winefordner, 1983). For the offline methodology, which is used for the instrument testing, it is important to define two different parameters: the instrument LOD and the method LOD. The instrument LOD was 1.3 nmol L$^{-1}$, determined as three times the standard deviation of the background when OF-UPW was injected into the sampling line. The method LOD was determined based on the reproducibility of the instrument background and the filter blanks. The reproducibility of the background was assessed by injecting several times different batches of OF-UPW. The value of 9 nmol L$^{-1}$, equivalent to three times the standard deviation of the resulting signals, was then used as a measure of this reproducibility and the offline method LOD. A similar LOD value was obtained as three times the standard deviation of the measurements of extracts of fractions of four different blank filters (2.2 cm$^2$) and was equal to 13 nmol L$^{-1}$ (for both quartz and teflon filters). We note that the average signal of these blanks was 25 nmol L$^{-1}$, which was subtracted from the signals measured when extracts of aerosol samples (with equivalent filter area) were injected.



### 3.1.2 Repeatability, reproducibility and response to selected model compounds

We assessed the instrument performance based on three repeated calibrations with 0, 30, 50, 100 and 150 nM $H_2O_2$ (Fig. S3). The instrument accuracy in determining the ROS concentration was found to be 3 % (n=15), based on the standard deviation of the slope of the linear fit. The instrument precision (repeatability) estimated at different $H_2O_2$ concentrations based on the fit prediction interval, was 25 %,

10 % and 5 %, at 30 nM, 70 nM and 150 nM, respectively. The instrument reproducibility was assessed based on the variation in the instrument sensitivity (in V $nM^{-1}$). In practice, we calculated the standard deviation of the response of 10 repeated measurements of known concentrations of $H_2O_2$ at different days using different WS. This reproducibility was found to be ~40 % (1 $\sigma$), which is much higher than the instrument precision, possibly due to the solution preparation and instrument operation conditions. Consequently, a calibration was always carried out at the beginning or at the end of each measurement series.

While the characterization tests discussed above were carried out using the offline mode, we obtained similar results when the instrument was used in the online mode. Fig. 3 shows that a similar linear relationship was obtained between the instrument response and the $H_2O_2$ concentration for the online (blue stars) and offline (red triangles) modes, resulting in statistically similar sensitivities (*t*-test, *p*-value = 0.93). This provides confidence in using the calibration and tests performed offline to predict online concentrations.

We also tested the response of the instrument to compounds expected to exhibit an oxidative potential, including peracetic acid (PAA;

Sigma-Aldrich, ~39 % in acetic acid, ≤ 6 % $H_2O_2$), tert-butyl hydroperoxide (tBuOOH; Aldrich, Luperox® TBH70X, 70 wt. % in water), benzoyl peroxide (BenP; Aldrich, Luperox® A75, 75 %, remainder water), lauroyl peroxide (LP; Aldrich, Luperox® LP, 97 %), tert-butyl peracetate (tBuPA; Sigma-Aldrich, Luperox® 7M50, 50 wt. % in aliphatic hydrocarbons), anthraquinone (AQ; Sigma-Aldrich, 97 %) and 2-butanone peroxide (2-BP, Sigma-Aldrich, Luperox® DHD-9, 32 wt. %). Table 1 provides an overview of the chemical structures of these compounds. The water soluble peroxides, i.e. PAA, tBuOOH and tBuPA, were dissolved in OF-UFW. The water insoluble compounds, i.e.,

BenP, LP and AQ were dissolved in ethyl acetate (Sigma-Aldrich, 99.8 %) and then diluted (by a factor of ~1000) using OF-UPW.

Response curves of the selected compounds with an expected oxidative potential compared to $H_2O_2$ are shown in Fig. 3. PAA showed a linear fluorescence intensity response similar to $H_2O_2$ (relative sensitivity *s* = 93 %). In contrast, AQ and organic peroxides like tBuPA barely reacted. Low responses were observed from tBuOOH (*s* = 25 %), BenP (*s* = 16 %) and LP (*s* = 15 %), as well as from 2-BP, which includes 3 -O-O- function groups (*s* = 21 %). The hydroperoxide groups in tBuPA, tBuOOH, BenP, LP and 2-BP are heavily protected by

tert-butyl, phenyl and alkyl groups which likely suppresses the reaction with DCFH. Less protected peroxides might be more reactive but such compounds are also less stable, and therefore not usually commercially available. This indicates that using a DCFH assay, the signal



intensity of peroxides varies significantly depending on the peroxide molecular structure and that sterically hindered peroxides may contribute much less to the DCFH signal.

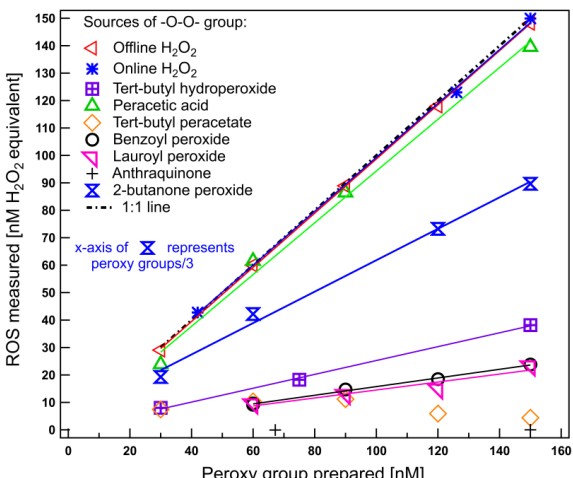

**Figure 3.** Calibration curves of $H_2O_2$ and response of selected compounds. Linear fits are shown for different peroxides and other compounds of interest in

5 the concentration range of 0 to 150 nM. The correlation coefficients $R^2$ were 0.99, except for lauroyl peroxide ($R^2 = 0.91$).

### 3.1.3 Instrument performance in ambient and smog chamber measurements

In order to evaluate the performance of the ROS analyzer in the field, two sets of experiments were conducted. In the first set, the instrument was operated in the offline mode using filter samples collected at two different sites: a) a site influenced by traffic emissions in Milano (Northern Italy), where quartz filters were sampled during October 2013 (Perrone et al., 2016) and b) a rural site in San Vittore (Southern

10 Switzerland in an Alpine valley) influenced by biomass burning, where samples were collected during January 2013 (Daellenbach et al., 2017; Zotter et al., 2014). The samples from both sites were stored in the freezer at -20 $^o$C for 2 years before analysis. A filter punch was dissolved in water and several samples were prepared by consecutive dilutions. Fig. 4 shows a linear relationship of the fluorescence response with increasing particle mass concentration (based on the mass on the filter punch and assuming 100 % water solubility) for both samples, where equivalent $H_2O_2$ concentrations span a wide range, which confirms the instrument linearity.



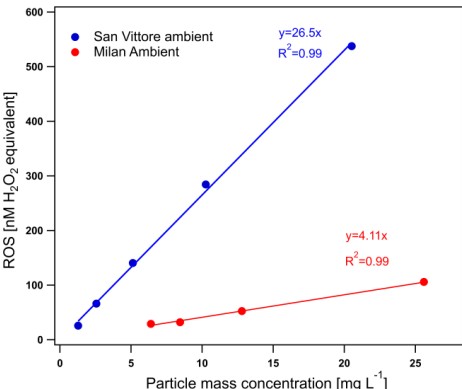

**Figure 4.** ROS content vs. dissolved particle mass concentration. Blue symbols represent $PM_{10}$ samples from San Vittore in winter (Switzerland), and red symbols TSP samples from Milan in autumn (Italy).

The second set of experiments was performed at the PSI smog chamber. Beech wood logs were combusted in a residential wood burner (Avant, 2009, Attika), following the procedure described in Bruns et al. (2016, 2017). The resulting emissions were sampled from the chimney through a heated line (473 K), diluted by a factor of ~8-10 using an ejector diluter (473 K, DI-1000, Dekati Ltd.) and injected into the smog chamber. Emissions were only sampled during the stable flaming phase for 11-21 min and the total dilution factors ranged from ~100 to 200. Experiments were conducted at -10 $^{\circ}$C or 15 $^{\circ}$C and at a relative humidity of ~50 %. After the characterization of the primary emissions, d9-butanol (butanol-D9, 98 %, Cambridge Isotope Laboratories) was injected into the chamber to determine the OH exposure from its decay (Barmet et al., 2012). A continuous injection of nitrous acid (2.3-2.6 L min$^{-1}$, ≥99.999 %) was used to create OH by photolysis. The chamber was then irradiated with UV light (40 lights, 90-100 W, Cleo Performance, Philips) for 4.5-6 h (Platt et al., 2013). Real-time characterization of the aerosols from the smog chamber was carried out throughout the experiment with the online ROS analyzer and a high resolution time of flight aerosol mass spectrometer (HR-ToF-AMS, Aerodyne Research).

The evolution of ROS measured by the online method is shown in Fig. 5 for one exemplary smog chamber aging experiment. Injection of the wood combustion emissions led to a primary organic aerosol (POA) concentration of 25 µg m$^{-3}$ and 26 nmol m$^{-3}$ of particulate ROS in the smog chamber. After the lights were switched on, secondary organic aerosol (SOA) was produced and total Organic aerosol (OA) measured by AMS reached a maximum concentration one hour later, but then decreased because of higher wall loss than the SOA production rate. The ROS concentration increased concurrently with the increasing OA, indicating the formation of ROS by photochemical reactions induced by OH radicals, but then decreased faster than OA. When we sampled through a particle filter inserted upstream of the



ROS online analyzer (pink areas), the ROS signal went to almost zero, which was considered as measurement base-line during aging (Fig. 5, Panel a).

To investigate the influence of aging on ROS formation, SOA and secondary ROS (ROS formed during aging, ROSs) were calculated by subtracting POA and primary ROS from the total OA and total ROS measured during lights on (Fig. 5, panel b), respectively. Here the POA

and primary ROS calculation was based on the assumption that they were not further oxidized after lights on and the wall loss rate was the same as for the inert tracer black carbon (BC). The content of ROSs in SOA (represented by ROSs/SOA) was in the range of 0.4-1.26 nmol $\mu g^{-1}$ within the oxidant OH exposure range of 0-30×$10^{-6}$ molec $m^{-3}$ h. Initially, aging resulted in a high ROSs content in SOA, which then decreased strongly with increasing OH exposure (Fig. 5, Panel c). This decrease could be due to further oxidation or decay of particulate ROS, indicating that first generation products from the VOCs oxidation might play a more important role in ROS formation than later

generation molecules.

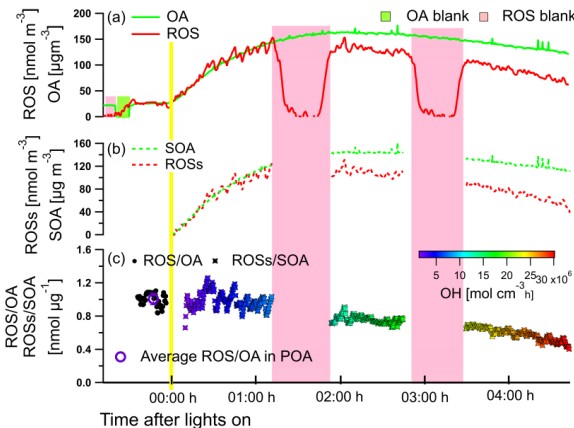

**Figure 5.** Evolution of the concentrations of OA mass and ROS during an online wood combustion smog chamber aging experiment. a) Total OA and ROS, b) SOA and ROSs, c) ROS content in the OA (before lights on) and ROSs content in the SOA (after lights on) as a function of the OH dose.

**3.2 Gas phase interference test**

We tested the potential interference of trace gases and aerosol components on the DCFH signal. In principle, at the applied sample flow rate, 99 % of the trace gases should get removed by the denuder. Specifically, we assessed the removal efficiency of the denuder with respect to the most abundant oxidizing trace gases $O_3$ and $NO_x$. After exposing the denuder to 464 ppb ozone for ~5 h, no increase in the background signal was observed (Table 2). Even without the denuder, 500 ppb NOx for 1 h showed no increase of the background signal. The results in





Table 2 indicate that a newly regenerated denuder completely removes NOx and O$_3$, making the denuder suitable for both smog chamber

(usually ~5 h aging per experiment) and ambient measurements (1 day/denuder replacement interval).

### 3.3 Particle phase matrix effects

### 3.3.1 Particulate SO$_4^{2-}$ and NO$_3^-$

Previous measurements of filters from Milano showed a clear correlation of ROS with the particulate SO$_4^{2-}$ and NO$_3^-$ concentration (Perrone

et al., 2016). During the investigated period, the average SO$_4^{2-}$ and NO$_3^-$ concentrations in Milan were 4 µg m$^{-3}$ and ~5-10 µg m$^{-3}$,

respectively. Here, we investigate whether SO$_4^{2-}$ and NO$_3^-$ exhibit a response in the DCFH-assay. Therefore, we tested the fluorescence

response to 23.5 µg m$^{-3}$ of SO$_4^{2-}$ (~5 times the ambient concentrations observed in Milan, prepared as (NH$_4$)$_2$SO$_4$) and to 228 µg m$^{-3}$ NO$_3^-$

(~30 times the ambient concentrations observed in Milan, prepared as NH$_4$NO$_3$), as well as the cross sensitivity of SO$_4^{2-}$ and NO$_3^-$ with H$_2$O$_2$

and 2-BP (Table 2).

Results show that the signals generated by injecting (NH$_4$)$_2$SO$_4$ and NH$_4$NO$_3$ were on average lower than the instrument background, which

were -4.8 and -3.5 nM (H$_2$O$_2$_e.q.), respectively. According to the reproducibility of the instrument background discussed in Sect. 3.1.1,

these differences are not statistically significant. For the cross sensitivity test, the fluorescence response of the SO$_4^{2-}$-H$_2$O$_2$ mixture (23.5 µg

m$^{-3}$ SO$_4^{2-}$ + 115 nM H$_2$O$_2$) and the NO$_3^-$-H$_2$O$_2$ mixture (228 µg m$^{-3}$ NO$_3^-$ + 115 nM H$_2$O$_2$) corresponded on average to 105 nM and 110 nM

H$_2$O$_2$ equivalent, respectively. These deviations from the value measured for H$_2$O$_2$ alone (115 nM) are not statistically different from zero

(Z-score test, $p$-value ~0.7 for SO$_4^{2-}$ and NO$_3^-$), within our measurement precision (Sect. 3.1.2). The SO$_4^{2-}$-2-BP mixture (23.5 µg m$^{-3}$ SO$_4^{2-}$ +

272.5 nM 2-BP) showed also a similar result. We conclude from these tests that particulate SO$_4^{2-}$ and NO$_3^-$, the most abundant single

particulate components, neither show any ROS signals nor influence the H$_2$O$_2$ and 2-BP measurements and that the observed relationship

between the secondary species and the ROS signals in ambient is rather a correlation and not based on causation.

### 3.3.2 Transition metals

Transition metals may induce a response through redox cycling. Iron is one of the most abundant transition metals in the aerosol (Valko et

al., 2005; Dall'Osto et al., 2016). However, potential iron-catalyzed ROS formation in an oxygen-rich environment has not yet been

examined using a DCFH assay. In order to investigate the effect of metals on the ROS signal we conducted two experiments: 1) the analysis

of the H$_2$O$_2$ reaction with DCFH in the presence of FeCl$_2$ (anhydrous, 99.998%, Sigma-Aldrich, USA) and FeCl$_3$ (FeCl$_3$·6H$_2$O,

Sigma-Aldrich, USA) and 2) the analysis of the H$_2$O$_2$ signal in the presence of ambient aerosols extracted from filter samples.



In the first set of experiments (shown in Fig. 6) the signal of $H_2O_2$ measured alone was compared with that of a mixed $FeCl_2$-$H_2O_2$ solution. At a concentration of 1 nM soluble $Fe^{2+}$ in water no influence on the ROS signal was observed within standard deviation. The same procedure was then applied to $H_2O_2$ (226 nM) combined with significantly higher $Fe^{2+}$ concentrations (182.5 nM). The fluorescence signals of the $Fe^{2+}$-$H_2O_2$ mixture, both with and without the presence of dissolved $O_2$, were significantly lower than the signal when measuring $H_2O_2$

5    alone. This might be due to the consumption of a substantial amount of $H_2O_2$ by $Fe^{2+}$, for the production of HO·, which will further react with $H_2O_2$ and resulting in further reduction of the $H_2O_2$ concentration (Kolthoff and Medalia, 1949). This indicates that concentrations of soluble $Fe^{2+} \leq 1$ nM, which were obtained at ambient concentrations of $\leq 10$ ng m$^{-3}$ soluble $Fe^{2+}$ in the online instrument, will not influence the ROS measurement. However, in cases of high ambient soluble $Fe^{2+}$ concentrations the ROS signal might be reduced, whereby this also depends on the $H_2O_2$ equivalent concentration. Measured ambient iron concentrations were found to be in the range of tens to several thousands of

10    ng m$^{-3}$ (Perrone et al., 2016; Oakes et al., 2012; Visser et al., 2015). Oakes et al. (2012) reported that water soluble Fe (II) constitutes between 2.5 and 32 % of total iron. Meanwhile, the $H_2O_2$-$Fe^{3+}$ mixture signal was observed to be almost the same as $H_2O_2$ signal alone with and without the presence of $O_2$, which is in agreement with the findings of LeBel et al. (1992) and Keenan et al. (2009). These findings were further evaluated below by examining the influence of genuine atmospheric particulate metals on the $H_2O_2$ signals.

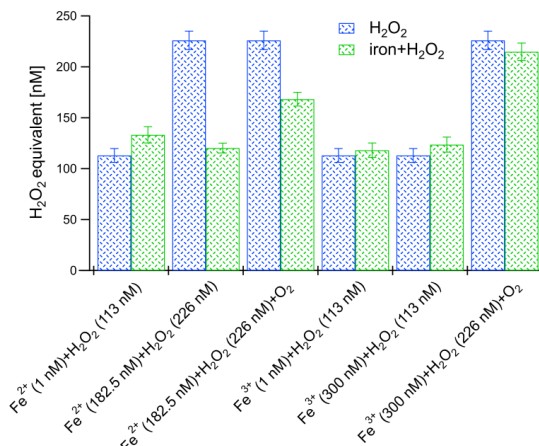

15    **Figure 6.** The relative fluorescence intensity during $Fe^{2+}$ and $Fe^{3+}$ cross sensitivity tests with $H_2O_2$. The blue bars represent the premixed $H_2O_2$ concentrations, and the green bars the [iron+$H_2O_2$] mixture concentrations. The error bars were calculated based on the instrument precision (see Sect. 3.1.2).





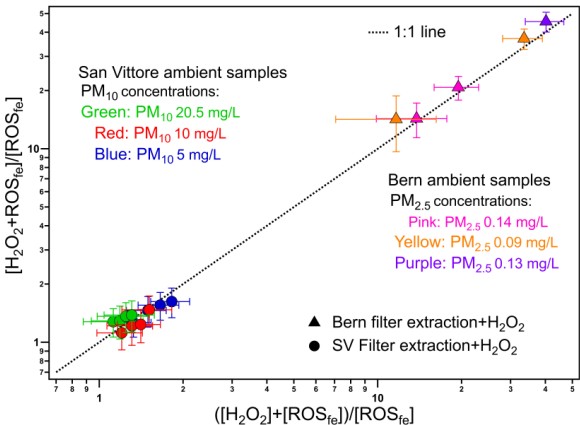

**Figure 7.** Comparison of the filter extract (fe)-$H_2O_2$ mixture with the sum of the separately measured filter extract and $H_2O_2$ response, both normalized to the filter extract signal. $[H_2O_2+ROS_{fe}]$ represents the fluorescence response of the filter extract-$H_2O_2$ mixture, $[H_2O_2]$ and $[ROS_{fe}]$ represent the fluorescence response of $H_2O_2$ and the filter extracts alone. Symbols represent different locations of the samples collected. Colors represent different PM

concentrations based on the mass on the filter punch and assuming 100 % water solubility. $H_2O_2$ concentrations mixed together with each PM concentrations ranged from 56.5-113 nM and 40-100 nM in Bern and San Vittore, respectively, which are also indicated indirectly on the x- and y-axes. Error bars represent the propagated uncertainty from the measurements of $[H_2O_2+ROS_{fe}]$, $[ROS_{fe}]$ and $[H_2O_2]$.

We then investigated whether the complex matrix of ambient particles, which also include different forms of iron together with other metals, have an influence on ROS measurements. For this second set of experiments, ambient filter samples from a rural site in San Vittore

(Switzerland) collected in January 2013 and an urban site located in Bern (Switzerland) collected in November 2014, were extracted and cross-tested with $H_2O_2$. In San Vittore, three concentrations of $PM_{10}$ from one filter punch were prepared; while in Bern, three concentrations of $PM_{2.5}$ from three different filters were prepared. Fig. 7 compares the fluorescence response of the filter extract-$H_2O_2$ mixture with the sum of the separately measured signals of the filter extract and of the $H_2O_2$. To account for the large differences in PM concentrations the signals were normalized to the signal of the filter extract. Results from both San Vittore and Bern lie on the 1:1 line

within our errors. This indicates that at ambient relevant concentrations the complex matrix of ambient particles has no influence on ROS signals.

**3.4 Assessment of ROS stability**

**3.4.1 Comparison of online and offline measurements**





A direct inter-comparison of online in-situ and offline filter sample measurements of the ROS content from different emission sources was performed. These aerosol samples included fresh and aged aerosols from wood combustion emissions from a smog chamber, as well as ambient aerosols collected in Bern (Switzerland).

The smog chamber experiments as well as the online performance were described in Sect. 3.1.3. In addition to the online measurements, the

particles from the chamber were collected on quartz filters (47 mm, Pall Corporation) at a flow rate of 26 L min$^{-1}$ for 30-32 min behind a charcoal denuder to remove organic vapors. Primary particles were collected after injection of the emissions into the smog chamber and before the lights were turned on. Aged particles were collected after around one and four hours of aging. The filters were immediately stored at 253 K and analyzed ~ 2 years after the smog chamber experiments.

Ambient measurements were performed at an urban site located at the Institute of Anatomy of the University of Bern. A stainless steel

cyclone (URG-2000-30ET, URG Corporation) was operated at a constant flow rate of ~ 100 L min$^{-1}$ to select particulate matter with an aerodynamic diameter <2.5 μm (PM$_{2.5}$). After size selection, particles were enriched using a versatile aerosol concentration enrichment system (VACES) (Kim, et al., 2001) and dried by passing through a diffusion dryer. Organic vapors were removed from the airstream using a charcoal denuder. The physico-chemical properties of the enriched aerosols were characterized using the online ROS analyzer, a scanning mobility particle sizer (SMPS, custom built) and a quadrupole aerosol chemical speciation monitor (ACSM, Aerodyne Research) for the

measurement of the non-refractory aerosol composition. Particles were collected up and down-stream of the VACES on Teflon filters (47 mm Fluoropore membrane, 3.0 μm pore size, Millipore, Molsheim, France) for quantification of particle-bound ROS. Prior to deposition on the filter, the sample flow was passed through a charcoal denuder removing oxidizing and organic gasses. Sampling time was 3 h and filters were immediately stored at -20 °C. Filter punches were then extracted as described in Sect. 2.1.3 and analyzed for the ROS content ~1 year after sampling.

The ROS concentrations measured by the online and offline method from the wood combustion experiments and ambient air in Bern are compared in Fig. 8. The ROS concentrations measured offline are on average 37 % lower than the online data in the Bern ambient measurements, and on average 67 % and 61 % lower than the online data for primary and secondary wood combustion samples, respectively. For the ambient measurements in Bern, a small number of measurements show agreement between the two methods indicating no ROS decay. A more detailed analysis is given in the following section to further explain the discrepancies of offline and online measurements.




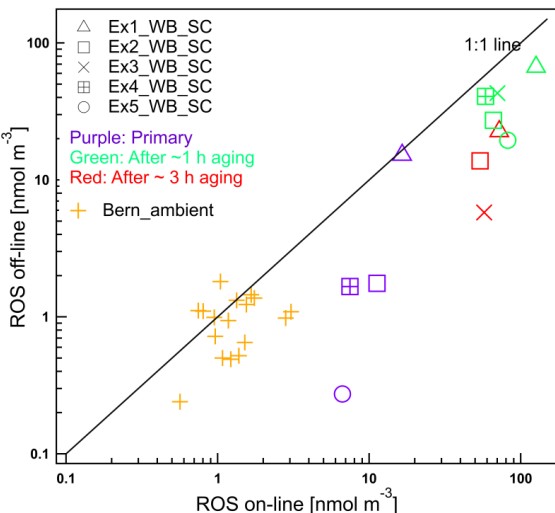

**Figure 8.** Comparison of online and offline measured ROS concentrations in the city of Bern in winter and during wood combustion smog chamber experiments (Ex*n*_WB_SC), including primary aerosol samples (purple), and secondary aerosol samples after aging for ~1 h (green) and ~4 h (red). A deviation from the 1:1 line indicates a discrepancy between the online and offline method. Filters from the wood combustion experiments were analyzed 2 years after sampling, and those from ambient measurements were measured 1 year later.

### 3.4.2 ROS degradation

As ROS decay with time, we investigated the evolution of the oxidation potential over time by measuring ROS from filter samples taken during additional biomass combustion laboratory experiments. The temperature of the filter samples was maintained at -20 $^{\circ}$C, except during transport which lasted ~3 h where the samples were packed at 0 $^{\circ}$C using ice packs. As this might have an additional effect on the results, ROS life-times determined at -20 $^{\circ}$C should be considered as the lowest estimates.

A pellet boiler was operated under two different conditions: high excess of combustion air ($\lambda^{++}$) and lack of combustion air ($\lambda^{-}$) (see Table 3). The emissions from the pellet boiler were sampled from the chimney through a heated line (473 K), and diluted by a factor of ~100-150 using two ejector diluters in series (VKL10, Palas GmbH). The emissions were then aged in a potential aerosol mass chamber (PAM) to simulate photochemical aging of the emissions and assess the potential of secondary organic aerosol (SOA) formation. The design and the use of the PAM chamber is described by Kang et al. (2007) and Bruns et al. (2015). Gas phase $O_2$ and CO (using a paramagnetic oxygen analyzer for $O_2$ and a non-dispersive infrared (NDIR) analyzer for CO, Ultramat 23 Siemens), $CO_2$ (NDIR analyzer, model LI-820, LI-COR®) as well as total volatile organic compounds and $CH_4$ (using a flame ionization detector (FID) with non-methane cutter, model 109A, J.U.M Engineering) were monitored in the hot, undiluted flue gas. In addition, non-methane volatile organic compounds (NMVOCs) as well as the OA, nitrate, ammonium and sulfate were measured after dilution using a proton transfer reaction-mass spectrometer (PTR-MS,





Ionicon) and a HR-ToF-AMS. Aerosol filter samples were taken for ~ 30 min on Teflon filters (47 mm Fluoropore membrane, 3.0 μm pore size, Millipore) after the PAM chamber for ROS offline analyses. The filters were stored in the freezer from hours up to four months before the measurements of the ROS-activity using the offline ROS setup (see Sect. 2.1.3).

The measured ROS concentrations in SOA from the different wood combustion experiments exhibit a clear decrease with increasing filter storage duration (Fig. 9). In addition, this decay seems to follow a double exponential function. This indicates the presence of a short-lived fraction $A_1$ with a decay constant $\pi_1 = \ln(2)/T_1$ and a long-lived fraction $A_2$ with a decay constant $\pi_2 = \ln(2)/T_2$, where $T_i$ represents the half-life. By constraining $A_2 = 1-A_1$; $0 \leq A_1, A_2 \leq 1$, a biexponential decay model was applied to fit the experimental values:

$$ROS_{norm}(t) = A_1 * EXP\left(-\pi_1 * (t - t_1)\right) + A_2 * EXP\left(-\pi_2 * (t - t_1)\right)$$                  Eq. (3)

where $t$ is the time after sampling, and $t_1$ is the time when the first offline measurement was performed. $ROS_{norm}(t)$ is the ROS measured at

time $t$ normalized to the ROS measured at time $t_1$. Measured and modeled values are compared in Fig. 9. The final modelling yields $\pi_1 = 0.0016$; $\pi_2 = 9.68$ resulting in a half-life of $t_1 \approx 1.7$ h for the short-lived ROS and $t_2 \approx 432$ days for the long-lived ROS.

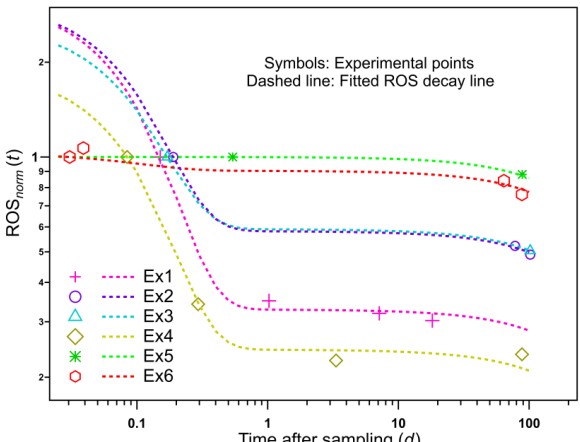

**Figure 9.** Measured and modeled ROS decays in SOA from wood combustion emissions with increasing sample storage duration for six experiments (Ex$n$). Symbols and dashed lines represent measured and modeled values, respectively. $ROS_{norm}(t)$ is the ROS measured at time $t$ normalized to the ROS

measured when the first offline was performed at time $t_1$. More information about the experiments can be found in Table 3. There is a very good agreement between measured and modelled (Fig. S4).

The total ROS concentration (in $H_2O_2$ equivalents) was dominated either by the short-lived or long-lived ROS fraction. We speculated that the abundance of the different ROS fractions might be influenced by the combustion conditions. Thus, the long-lived fraction of ROS was




compared with various wood combustion parameters. No correlation was found with $\lambda$, CO, $CO_2$ and NMVOCs (defined in Table 3), nor specific gas-phase families, e.g., PAHs, furans, oxygenated aromatics, N-containing or O-containing compounds. However, as shown in Fig. S5 the fraction of long-lived ROS seems to be negatively correlated with the modified combustion efficiency (MCE) and the total OA mass present in the chamber (with Ex4 as an exception). These results might indicate that the composition of ROS formed during wood

combustion depends on the combustion conditions. As semi-volatile organic compounds have a higher chance to condense to the particle phase with increasing OA concentration, the anti-correlation of the long-lived fraction of ROS with OA concentration suggests that the more oxidized/low volatility ROS tend to have longer life-times than the less oxidized/higher volatility ROS.

Estimations of ROS lifetimes were done previously. ROS species measured in oxidized oleic acid particles was separated into short- and long-lived species with a half-life of a few minutes and hours to days, respectively (Fuller et al., 2014). Chen et al. (2011) determined a ROS

half-life of 6.5 h in oxidized organic aerosols. Krapf et al. (2016) showed that more than 60 % of peroxides contained in SOA from α-pinene ozonolysis decayed with a short half-life of 45 min.

To compare the ROS online measurement with immediate offline measurements, 2,6-dimethoxyphenol was used as a precursor and aged in the PAM chamber. SOA was then sampled on a Teflon filter (47 mm Fluoropore Membrane, 3.0 μm pore size, Millipore), at a flow rate of 1.7 L min$^{-1}$ for ~1 hour after passing through a similar charcoal denuder as applied for the online measurements. The filter was then

measured directly after sampling. Results showed that the offline measurement was 40 % lower than the online measurement indicating that already without significant sample storage duration the short-lived ROS fraction was lost in the offline methodology. This is in agreement with Fuller et al. (2014) and Krapf et al. (2016) who showed that a larger fraction of ROS in fresh SOA decays within tens of minutes.

A summary of the ROS decay behavior in aerosols from Bern ambient and wood combustion experiments, a normalized frequency distribution of the ROS decay percentage of different sources is plotted in Fig. S6. The decay percentage of ROS was calculated as follows:

$$ROS_{decay\ percentage} = \left[\frac{ROS_{online} - ROS_{offline}}{ROS_{online}}\right] \times 100\ \%$$           Eq. (4)

The normalized frequency of a specified ROS decay percentage, was then calculated as the ratio of the number of experiments yielding a certain decay percentage normalized to the number of total experiments. From Fig. S6 we conclude that the most frequently occurring ROS decay percentages were 40-80 % in wood combustion experiments, whereby aging in the smog chamber and PAM chamber yielded similar results. Similarly, around 60 % of ROS decayed in the majority of all the 27 ambient samples collected in Bern. Overall, the offline method

underestimates the ROS content due to the degradation of short-lived ROS species prior to filter analysis. The comparison of online and



offline ROS measurements from ambient and wood combustion smog chamber experiments indicates that on average $60 \pm 20$ % of ROS decayed during filter storage and handling, highlighting the importance of online measurements.

**4 Conclusions**

In this study, a modified online and offline ROS analyzer was presented and characterized. The major improvements to optimize the

analysis were: 1) degassing of the water and PBS to prepare the working solutions; 2) separation of DCFH and peroxidase working solutions, which were then mixed just before reaction with the sample solution; 3) no ultrasonic filter extraction for offline analysis. All these efforts resulted in an instrument LOD of 2 nmol m$^{-3}$ and 1.3 nmol L$^{-1}$ for online and offline analysis, respectively. The method LOD of the offline analysis was higher, with 9 nmol L$^{-1}$ and 13 nmol L$^{-1}$ based on the variability of the instrument background and the filter blanks, respectively. The instrument accuracy in determining the ROS concentration was found to be 3 %, and the instrument precision (repeatability) was 25 %,

10 % and 5 %, at 30 nM, 70 nM and 150 nM, respectively. The reproducibility of the instrument sensitivity was ~40 % due to solution preparation and instrument operation, thus a calibration is needed for each experiment and new batch of WS.

As shown with model organic compounds only peroxyacids were quantitatively measured, while large organic peroxides or those with bulky functional groups (i.e., tert-butyl and phenyl) strongly reduced the fluorescence response of the DCFH-assay. Potential interferences from gas phase $O_3$ and $NO_x$ were not observed and matrix effects of particulate $SO_4^{2-}$ and $NO_3^-$ were not statistically significant. While $Fe^{3+}$ does

not show a detectable interference, high soluble $Fe^{2+}$ concentrations present in ambient aerosol could reduce the ROS signal.

Both online and offline measurements with the analyzer were performed in field and laboratory experiments. ROS concentrations from offline field measurements showed a linear relationship with increasing ambient particle concentrations. Smog chamber aging experiments of wood combustion emissions revealed a high initial ROS content in SOA, which then strongly decreased with OH-exposure. Generally, ROS decayed with increasing filter storage duration. Due to the degradation of the highly reactive ROS, the offline method generally

underestimates the ROS concentration, on average by $60 \pm 20$ %. From the decay behavior, ROS in SOA can be separated into two categories: a short-lived/highly reactive fraction with a half-life of ~1.7 h and long-lived/less reactive species with a half-life of ~432 d. Consequently, to obtain a better estimate of the real ROS concentration in the ambient air or in simulation chamber experiments, a fast online method as presented in this study is advantageous.

**Acknowledgements**





This study was financially supported by the Swiss National Science Foundation (NRP 70 "Energy Turnaround") and the China Scholarship Council (CSC) under grant agreement no. 201007040040. The research leading to these results also received funding from the European Community's Seventh Framework Programme (FP7/2007-2013) under grant agreement no. 290605 (PSI-FELLOW) and from the Competence Center Environment and Sustainability (CCES) (project OPTIWARES). The authors thank M. G. Perrone and M. Krapf for

providing the ambient filters, M. Xiao for the helpful discussions, R. Richter and G. Wehrle for their competent technical advice, as well as S. Browns, I. Gavarini, L. E. Cassagnes and D. Bhattu for their support in the lab.

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



**Table 1.** Model organic peroxides used in this study.

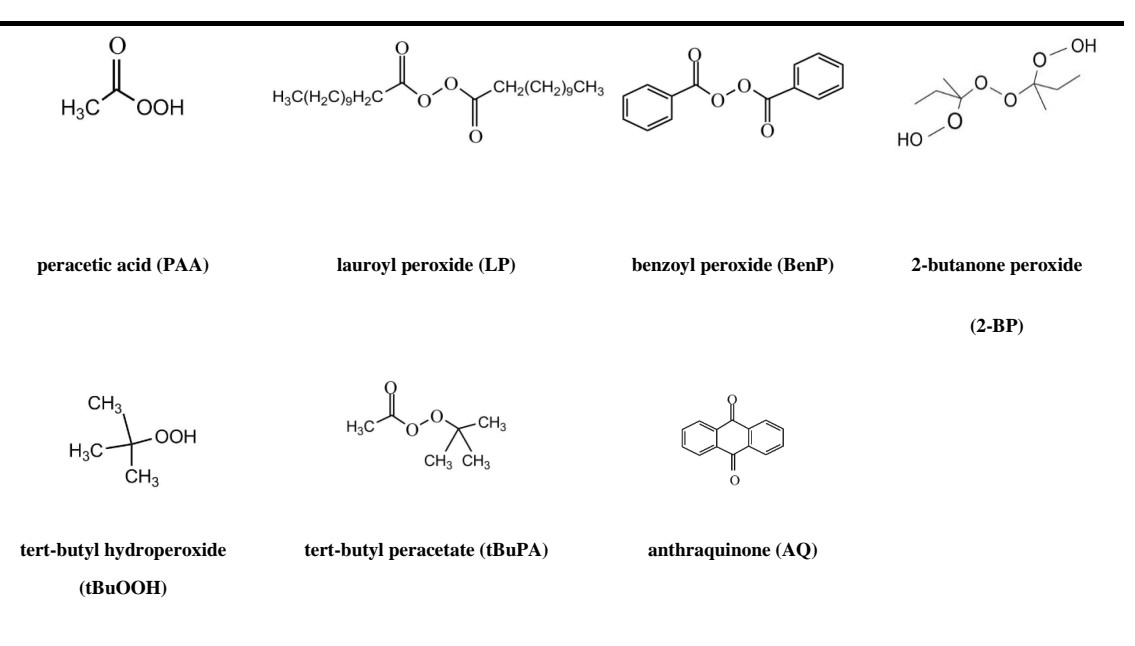

| peracetic acid (PAA) | lauroyl peroxide (LP) | benzoyl peroxide (BenP) | 2-butanone peroxide (2-BP) |

| tert-butyl hydroperoxide (tBuOOH) | tert-butyl peracetate (tBuPA) | anthraquinone (AQ) |



**Table 2.** Effects of the potential interferences in the gas and aerosol phase on the DCFH signal.

| | Species tested | Concentration applied | | Measured concentration ($H_2O_2$_eq.) | |
|---|---|---|---|---|---|
| | | without denuder | with denuder | without denuder | with denuder |
| Gas phase | $O_3$ | 464 ppb | 464 ppb[*] | 150 nM | 0 nM |
| | $NO_x$ | 0-500 ppb | 0 nM | 0 nM | 0 nM |
| Particle phase | $SO_4^{2-}$ | 23.5 µg m$^{-3}$ | - | -4.8 nM | - |
| | $NO_3^-$ | 228 µg m$^{-3}$ | - | -3.5 nM | - |
| | $SO_4^{2-}$+$H_2O_2$ | 23.5 µg m$^{-3}$+ 115 nM | - | 105 nM | - |
| | $NO_3^-$+$H_2O_2$ | 228 µg m$^{-3}$+ 115 nM | - | 110 nM | - |
| | $SO_4^{2-}$+2-BP | 23.5 µg m$^{-3}$+ 272.5 nM | - | 272.5 nM | - |

* Denuder was exposed for ~ 5 h



**Table 3.** Short-lived and long-lived ROS fractions and parameters from the different experiments (Ex$n$=denotes the number of the experiment).

| Filter | Ex1 | Ex2 | Ex3 | Ex4 | Ex5 | Ex6 |
|---|---|---|---|---|---|---|
| $\lambda$[#] | 1.31 ($\lambda^-$) | 3.25 ($\lambda^{++}$) | 3.33 ($\lambda^{++}$) | 3.18 ($\lambda^{++}$) | 3.16 ($\lambda^{++}$) | 3.36 ($\lambda^{++}$) |
| MCE[**] | 0.99 | 0.98 | 0.97 | 0.98 | 0.98 | 0.96 |
| T (Chamber, °C) | 37.9 | 37.9 | 37.9 | 39.8 | 39.8 | 39.8 |
| RH (Chamber, %) | 18.6 | 24 | 24.5 | 20.9 | 20.9 | 20.9 |
| OA [$\mu g\ m^{-3}$][##] | 43.0 | 39.1 | 29.0 | 4.5 | 9.9 | 16.5 |
| $CH_4$ (ppmv, norm[*])[&] | 0.017 | 0.16 | 0.16 | 0.027 | 0.087 | 0.13 |
| CO (ppmv, norm[*])[&] | 2.2 | 11.0 | 11.5 | 4.5 | 6.3 | 8.6 |
| $CO_2$ (ppmv, norm[*])[###&] | 375.5 | 391.5 | 381.1 | 210.8 | 212.13 | 203.0 |
| NMVOCs (ppm, norm[*])[&] | 0.04 | 0.74 | 0.78 | 0.13 | 0.45 | 0.6 |
| Long-lived fraction ($A_1$) | 29.0 % | 53.7 % | 70.1 % | 18.8 % | 100 % | 97.5 % |
| Short-lived fraction ($A_2$) | 71.0 % | 46.3 % | 29.9 % | 81.2 % | 0 % | 2.5 % |

[#]air fuel equivalence ratio ($\lambda$). $\lambda = O_{2,amb}[\%]/(O_{2,amb}[\%] - O_{2,exh}[\%])$ where $O_{2,amb}$ and $O_{2,exh}$ are the oxygen contents in ambient air ($O_{2,amb}$ = 21 %) and the one measured in the flue gas, respectively.

[##] OA = primary OA + secondary OA

[###] background corrected values

[&]all the concentrations of gas and particle phase compounds are after the PAM.

[*]norm indicates that concentrations are reported at 0 °C and 1013 mbar and normalized to a reference $O_2$ content of 13 %, $x_{norm}$ = [species $x$] × $\lambda_{actual}/\lambda_{reference}$.

[**]MCE = [$CO_2$]/([$CO_2$] + [CO]) (Ward and Radke, 1993).

[***] Non-methane VOCs (NMVOCs) = VOC-$CH_4$

