# Peer review of "Development, characterization and first deployment of an improved online reactive oxygen species analyzer"

_Atmospheric Measurement Techniques, 2017_

## Referee Comment (RC1) · Anonymous Referee #1 · 18 Jul 2017

The paper from Zhou et al., describes the development of a new online system for measuring the ROS concentration of PM. The paper makes a comparison between the newly developed online system and conventional offline system for ROS measurement. It also highlights the advantages of using an online system for measuring ROS concentration and the negative effects of long filter storage time on the offline ROS measurements. The authors then put the instrument in the field and used it many campaigns, which demonstrate the workability of the instrument. From reading the paper, it seems to me that the authors are confused with their terminology. They have been using the term oxidative potential and ROS interchangeably, which are actually different. Oxidative potential is the capability of the PM to generate the reactivee oxygen

species. What the authors are measuring is the ROS, which is already in the particulate phase. The species like quinones, metals can generate the ROS in a suitable reductive environment so they could considered as the precursors of ROS, but the author's instrument specifically measure the particulate ROS and not the potential of such species to generate ROS. In that sense, I think the experiments conducted using Fe are meaningless. Another concern is that the authors have not talked anywhere about the collection efficiency of their PILS. If they have done these tests in previous papers then they should report those figures again in this paper. The reviewer feels that more details regarding the online instrument operation procedure and field set up needs to be added.

Some of the specific comments are below:

Page 3, lines 10-15: It was mentioned that PILS is also called mist chamber. PILS is generally used to refer to a very specific aerosol collection device (Orsini et al.,2003). Whereas, mist chamber aerosol collector is usually used when referring to Cofer Scrubber (or mist chamber) (King et al.,2013). The device mentioned in this study is also an aerosol particle collector (Takeuchi et al., 2005), however the nomenclature used in this paper could cause some confusion for the readers. Hence, instead of referring as PILS it would be better to refer to it as aerosol/particle collector. Page 10: Lines 5-10: The air stream 1.7LPM was mixed with OF-UPW and sprayed into the mist chamber with 0.3 ml/min. Was there any loss in volume of OF-UPW which was filtered from the hydrophilic filter? What was the volume of extract used for ROS concentration analysis? Was the same volume used for both online and offline analysis? The ambient air sampling duration adopted for the online system and the minimum sampling time required to get a ROS concentration which is above the detection level of the online system, should also be mentioned. Regarding the hydrophilic and hydrophobic used in this study, how frequently was it required to replace them? Was there clogging of filter pores (ie. pressure drop) which could affect the air flow. More details about the daily maintenances (frequency of filter replacements, frequency of replacing the solvents

etc.) and discussion about the portability of the system can be included. Some more discussion about the extent of automation of the system (was all the online system experiments described in this paper performed without any manual assistance?) would be beneficial.

References Takeuchi, M., Ullah, S. R., Dasgupta, P. K., Collins, D. R., & Williams, A. (2005). Continuous collection of soluble atmospheric particles with a wetted hydrophilic filter. Analytical chemistry, 77(24), 8031-8040. Orsini, D. A., Ma, Y., Sullivan, A., Sierau, B., Baumann, K., & Weber, R. J. (2003). Refinements to the particle-into-liquid sampler (PILS) for ground and airborne measurements of water soluble aerosol composition. Atmospheric Environment, 37(9), 1243-1259. King, L. E., & Weber, R. J. (2013). Development and testing of an online method to measure ambient fine particulate reactive oxygen species (ROS) based on the 2', 7'-dichlorofluorescin (DCFH) assay. Atmospheric Measurement Techniques, 6(7), 1647.

---

## Referee Comment (RC2) · Anonymous Referee #2 · 1 Aug 2017

This manuscript described the design and characterisation of an online ROS analysis systems very similar to instruments built by other groups before. A number of aspects are not described in sufficient detail and the following points need to be considered before publication.

p.1, line 14/15 (Abstract): it is unclear what the detection limit for offline is. 1.3 or 9-13nmol L-1. Also, indicate with what compound this detection limit was determined. nmol H2O2 L-1?

p.2, line 15: Please also reference Wang et al., Journal of Toxicology, 2011, who first developed an online DCFH system.

[Figure]

p.3, line 12: What is the difference between reaction and incubation?

p.3, line 15/16: For how long was the denuder efficient and how was this assessed? The HRP assay is sensitive to H2O2. Was the denuder efficient in removing gaseous H2O2? How was that determined?

p.4, line 12/13. Please support the statement in this sentence with evidence. By how much was the lifespan of the solution shortened and how did the additional contamination affect the measurement?

p. 7, section 2.2 I am not sure this section is necessary as it does not add any information.

p.8, line 1: With what experiment was the residence and response time determined.

Fig. 2: How is the difference in detector response reconciled between the two compounds?

p.8, line 8: Was the LOD online determined by using ambient air?

p.9 & Fig.S3: Looking at the data in Fig.S3 and confidence intervals shown it look to me the detection limit is more in the range of 15nM. How does that compare with numbers discussed on p.8.

p.9, line 20: Did the use of ethyl acetate affect the reactivity of HRP and/or DCFH? How was this verified? The enzyme HRP could be strongly impaired in its reactivity in an organic solvent.

p. 10, line 5: Data are only shown from 30-150nM, not 0 – 150nM.

p. 11, Fig. 4: No detail is given for the data shown in Fig. 4. What are the dates, collected? Do they correspond to data shown in the references cited? Why is there a difference in the two data sets, what are the errors on the data shown etc.

p.13, line 14: SO4 and NO3 (given in units of ug m-3) were mixed with H2O2 (given in

units of nM) for cross sensitivity test. How was that done. Was the sulfate and nitrate nebulised as aerosol? A lot more detail needs to added here.

p. 14/Fig. 6: The experiments with Fe2+/Fe3+ are discussed in a purely descriptive way. A more detail discussion rationalising the results and referencing Fenton reactions is needed.

p.14, line 11: It is mentioned that water-soluble Fe2+ is measured up to 100s ng m-3 in ambient samples. To what concentration does that correspond in the working solution. Could the ROS signal potentially be suppressed under these conditions? This should be discussed more clearly.

p. 17/Fig. 8: Are the units for the x and y axis "nmol H2O2 m-3"? If yes, this should be indicated explicitly.

p.16, line 15/Fig.8: It is mentioned that filters were collected before and after the VACES. Was ROS different in the filters collected before and after the VACES? Did the use of a VACES affect online ROS concentrations?

Fig 9 and related discussion: It is not acceptable to derive a half-life of ROS from the data shown in Fig. 9. The time resolution of the data presented is far too sparse to constrain the half-life of ROS to any reasonable accuracy. This Figure has to be deleted or much more data has to be provided to make a meaningful statement about ROS half-life.

p. 20, line 4: It should be explained to what "improvements" compares to. Similar instruments by other groups use some of the "improved" conditions as well. Please me precise in your statements.

p. 20. Line 21: See above. The data presented here cannot support a lifetime estimate.

---

## Author Comment (AC1) · 30 Oct 2017

We thank the referees for their insightful questions and comments, which helped improving the quality of the paper. Our answers are listed in the following in red, after the reviewer's comments, which are in black. The modifications in the text are marked in yellow.

**Anonymous Referee #1**

General comments:

The paper from Zhou et al., describes the development of a new online system for measuring the ROS concentration of PM. The paper makes a comparison between the newly developed online system and conventional offline system for ROS measurement. It also highlights the advantages of using an online system for measuring ROS concentration and the negative effects of long filter storage time on the offline ROS measurements. The authors then put the instrument in the field and used it many campaigns, which demonstrate the workability of the instrument. From reading the paper, it seems to me that the authors are confused with their terminology. They have been using the term oxidative potential and ROS interchangeably, which are actually different. Oxidative potential is the capability of the PM to generate the reactive oxygen species. What the authors are measuring is the ROS, which is already in the particulate phase. The species like quinones, metals can generate the ROS in a suitable reductive environment so they could considered as the precursors of ROS, but the author's instrument specifically measure the particulate ROS and not the potential of such species to generate ROS. In that sense, I think the experiments conducted using Fe are meaningless.

In the literature, the oxidative potential (OP) is considered as a measure of the capacity of PM to oxidize target molecules (Janssen et al., 2014). This can happen by the capability of (a) particle borne components to act as reactive oxygen species (ROS) or (b) of particle borne components to mediate ROS formation in the target environment. Acellular chemical assays try to quantify either one or both of these effects. However, all assays have their limitations and do not provide a full answer to either of the two oxidation processes. We agree with the reviewer that the DCFH assay targets mostly point (a), by measuring the capacity of the PM components to oxidize 2',7'-dichlorofluorescin (DCFH) to the fluorescent compound dichlorofluorescein (DCF) in the presence of horseradish peroxidase (HRP). We do show that the efficiency of the reaction of DCFH to oxidants/peroxides varies substantially and that components known to induce redox cycling (e.g. metals and quinones) do not seem to react with DCFH. In the corrected version of the manuscript we highlighted the point raised by the reviewer that DCFH measures the capability of particle borne components to act as reactive oxygen species, rather than the potential of the components to mediate ROS formation. The related modifications in the new manuscript are as follows:

Page 8: 2) Response of the DCFH assay to selected components with expected capability to act as reactive oxygen species (Sect. 3.1.2 and 3.3.2).

Page 10: We also tested the response of the instrument to compounds expected to exhibit the capability to act as reactive oxygen species,

Page 10: Response curves of the selected compounds with an expected capability to act as reactive oxygen species compared to $H_2O_2$ are shown in Fig. 3.

**Page 11:** Also components known to induce redox cycling (e.g. metal ions and anthraquinone) do not seem to react with DCFH. Thus we conclude that DCFH measures the capability of particle borne components to act as reactive oxygen species, rather than potential of species to mediate ROS formation.

Our main goal of conducting the experiments with Fe and anthraquinone was to clarify potential matrix effects of these components to the ROS signal. Iron exists widely in different emission sources as well as in ambient particles (Dall'Osto et al., 2016; Valko et al., 2005). Water soluble Fe (II) exists in genuine atmospheric particulate matter (Oakes et al., 2012), which might react with the pre-existing ROS on particles thus influencing the ROS signal. We further evaluated the matrix effects of the genuine atmospheric particulate matter (as it is supposed to contain Fe (II)) to the $H_2O_2$ signals in the same Section (Sect. 3.3.2). We think these tests are indeed required.

Another concern is that the authors have not talked anywhere about the collection efficiency of their PILS. If they have done these tests in previous papers then they should report those figures again in this paper.

Our particle collector was constructed according to Takeuchi et al. (2005). These authors did an extensive characterization of the collection efficiency. For the combination of a hydrophilic cellulose filter, supported by a 5.0 µm pore size hydrophobic membrane filter they determined a collection efficiency for water-soluble particles of 80 % for 100 nm particles and higher than 97.7% for particles > 280 nm. This information was now added in the revised manuscript on Page 4.

Added text: The collection efficiency for water-soluble particles was determined by Takeuchi et al. (2005) to be 80 % for 100 nm particles and higher than 97.7 % for particles > 280 nm.

The reviewer feels that more details regarding the online instrument operation procedure and field set up needs to be added.

Based on the specific comments below and by the second reviewer, more specific information has been added in Sect. 2.2:

2.2 Instrument maintenance and portability

The instrument can be easily disassembled and rebuilt to be used in both laboratory and field campaigns. The instrument is not yet fully automatized. The following manual operations are required: 1) calibration; 2) replacing the hydrophilic and hydrophobic filters in the aerosol collector and the denuder every 2-3 days during ambient measurements; while in laboratory experiments, we exchanged the denuder for each laboratory experiment (~ 5 h) to be on the safe side; 3) regularly switching the air inlet channel to the particle-free mode (ROS blank) and checking the air flow during the measurement (before the experiment, during the experiment and after the experiment) to insure that the air sample flow was constant at 1.7 L min$^{-1}$; 4) cleaning of the ROS analyzer with 1 M $H_2SO_4$ for ~ 12 hours every two weeks to remove contaminations in the system; 5) replacing all the tubes used in the system every half year.

Specific comments:

Page 3, lines 10-15: It was mentioned that PILS is also called mist chamber. PILS is generally used to refer to a very specific aerosol collection device (Orsini et al.,2003). Whereas, mist chamber aerosol collector is usually used when referring to Cofer Scrubber (or mist chamber) (King et al., 2013). The device mentioned in this study is also an aerosol particle collector (Takeuchi et al., 2005), however the nomenclature used in this paper could cause some confusion for the readers. Hence, instead of referring as PILS it would be better to refer to it as aerosol/particle collector.

Indeed, the main part of our aerosol collector is a "mist chamber". We use now the term "aerosol collector" throughout the manuscript (as the design is based on Takeuchi et al., 2005).

Page 10: Lines 5-10: The air stream 1.7 LPM was mixed with OF-UPW and sprayed into the mist chamber with 0.3 ml/min. Was there any loss in volume of OF-UPW which was filtered from the hydrophilic filter?

We think here you are referring to Page 4, Lines 5-10: Yes, there is a potential loss in the volume of the OF-UPW. The air leaving the mist chamber will be saturated by water by taking up some water depending on the humidity of the sample air. If we assume a dry sample air the uptake of water would be 10 %. Since we do our calibrations with filtered ambient air, we do not introduce a large error by not considering RH of the sample air in our calculations.

What was the volume of extract used for ROS concentration analysis?

The OF-UPW is continuously injected at 0.3 mL min$^{-1}$ together with an aerosol sample flow of 1.7 L min$^{-1}$. The full aerosol extract (minus a small fraction lost due to saturation of the air as mentioned above) is pumped out of the aerosol collector and mixed with 0.4 mL min$^{-1}$ of working solution. This resulted in a total flow rate of 0.7 mL min$^{-1}$.

We modified the description to (page 4): The 1.7 L min$^{-1}$ air stream was mixed with the OF-UPW, which was continuously sprayed into the mist chamber with a flow rate of 0.3 mL min$^{-1}$, where the aerosol particles were incorporated into the water droplets. The liquid containing the water soluble fraction of the aerosol was collected at the bottom of the aerosol collector at a flow rate of 0.3 mL min$^{-1}$ and then mixed with the working solution at a flow rate of 0.4 mL min$^{-1}$ for analysis. This resulted in a total flow rate of 0.7 mL min$^{-1}$. Therefore, the measurement of ROS is continuous, which provides real time measurement of ROS.

Was the same volume used for both online and offline analysis?

In general, we extracted a filter punch of 14 mm Ø in 10 mL of OF-UPW. However, the filter area and/or the volume of the OF-UPW was sometimes adjusted to keep the extracted ROS concentration in the measurement range of the instrument. The extract was then injected into the ROS analyzer in the same way as the online method with a rate of 0.3 mL min$^{-1}$ and mixed with the working solution at a rate of 0.4 mL min$^{-1}$ for analysis. Thus the mixture of the flows was the same for online and offline analysis.

The text was added for clarification to the manuscript (page 5): In general, we extracted a filter punch of 14 mm Ø of the filter area in 10 mL of OF-UPW for 15 min at 30 °C. However, the filter area and/or the volume of the OF-UPW was sometimes adjusted to keep the extracted ROS concentration in the measurement range of the instrument. The vial was then vortexed (Vortex Genie 2, Bender& Holbein AG, Switzerland) for 1 min to ensure homogeneity and filtered through a 0.45 μm nylon membrane syringe filter

The ambient air sampling duration adopted for the online system and the minimum sampling time required to get a ROS concentration which is above the detection level of the online system, should also be mentioned.

The instrument is measuring continuously. This is different from the method described in King and Weber (2013). We do not collect a certain amount of sample before the analysis is taking place. Basically our instrument collects 1.7 Liter air into a total solution of 0.7 ml for the analysis. We have a ROS detection limit of 2 nmol m$^{-3}$ of sampled air as described in the manuscript.

Regarding the hydrophilic and hydrophobic used in this study, how frequently was it required to replace them? Was there clogging of filter pores (i.e. pressure drop) which could affect the air flow. More details about the daily maintenances (frequency of filter replacements, frequency of replacing the solvents etc.) and discussion about the portability of the system can be included. Some more discussion about the extent of automation of the system (was all the online system experiments described in this paper performed without any manual assistance?) would be beneficial.

We checked the air flow during the measurement regularly (before, during and after the experiment) to insure that the air sample flow was constant at 1.7 L min$^{-1}$. In case of the laboratory experiments we changed the hydrophilic and hydrophobic filters before each experiment to avoid pollutant interferences from the previous experiments as well as clogging of the filter pores. During ambient measurements we changed the hydrophilic and hydrophobic filters every 2-3 days. This does not mean that it was necessary to change it that frequently. It was more done so as a precaution. We did not systematically investigate this issue.
Usually we prepared the OF-UPW, which lasted for several days of continuous measurements. We did not observe an increase of background signal during this time. The instrument is not yet fully automatized. We needed to manually switch the aerosol inlet to the particle free inlet when performing the ROS blank measurements.
We added the description of the particle free inlet on page 14: Further, we regularly checked the ROS blank by measuring particle-free air by switching a 3-port valve and sampling through a particle filter (disposable filter units, Balston, UK) installed in another line.

Some other parts like the denuder, the hydrophilic and hydrophobic filters used in the aerosol collector need to be manually changed accordingly. We added the details of instrument maintenance and portability on page 7, Sect. 2.2, which we have mentioned in the previous general comments.

References:

Dall'Osto, M., Beddows, D. C. S., Harrison, R. M., and Onat, B.: Fine Iron Aerosols Are Internally Mixed with Nitrate in the Urban European Atmosphere, Environmental Science & Technology, 50, 4212-4220, 2016.

Janssen, N. A. H., Yang, A., Strak, M., Steenhof, M., Hellack, B., Gerlofs-Nijland, M. E., Kuhlbusch, T., Kelly, F., Harrison, R., Brunekreef, B., Hoek, G., and Cassee, F.: Oxidative potential of particulate matter collected at sites with different source characteristics, Science of The Total Environment, 472, 572-581, 2014.

Oakes, M., Weber, R. J., Lai, B., Russell, A., and Ingall, E. D.: Characterization of iron speciation in urban and rural single particles using XANES spectroscopy and micro X-ray fluorescence measurements: investigating the relationship between speciation and fractional iron solubility, Atmos. Chem. Phys., 12, 745-756, 2012.

Valko, M., Morris, H., and Cronin, M. T. D.: Metals, Toxicity and Oxidative Stress, Current Medicial chemistry, 12, 1161-1208, 2005.

---

## Author Comment (AC2) · 30 Oct 2017

We thank the referees for their insightful questions and comments, which helped improving the quality of the paper. Our answers to referee #2 are listed in the following in red, after the reviewer's comments, which are in black. The modifications in the text are marked in yellow.

**Anonymous Referee #2**

This manuscript described the design and characterization of an online ROS analysis systems very similar to instruments built by other groups before. A number of aspects are not described in sufficient detail and the following points need to be considered before publication.

p.1, line 14/15 (Abstract): it is unclear what the detection limit for offline is. 1.3 or 9-13nmol L-1. Also, indicate with what compound this detection limit was determined. nmol H2O2 L-1?

We report two types of detection limits, the instrument detection limit and the method detection limit. The details of the definition and their determination are given in Sect. 3.1.1. The instrument LOD was 1.3 nmol $L^{-1}$, determined as three times the standard deviation of the background when OF-UPW was injected into the sampling line. The method LOD was determined based on the reproducibility of the instrument background and the filter blanks. Due to the varying background of the offline instrument between different samples and the variation in filter blanks and filter extraction a higher method detection limit was found. Due to another question raised by this referee we re-evaluated the method detection limit (see below). Instead of 9-13 nmol $L^{-1}$ we give now 18 nmol $L^{-1}$ as the highest determined method detection limit. Our instrument is calibrated with $H_2O_2$.
We added in the abstract that the instrument is calibrated with $H_2O_2$.

p.2, line 15: Please also reference Wang et al., Journal of Toxicology, 2011, who first developed an online DCFH system.

We added this reference.

p.3, line 12: What is the difference between reaction and incubation?

We used it in the way that incubation relates to the experimental conditions and reaction to the chemical process under those conditions. As incubation is a well-defined process in biology and medicine and does not conform to our use, we changed the terminology. We replace incubation by reaction throughout the manuscript.

p.3, line 15/16: For how long was the denuder efficient and how was this assessed? The HRP assay is sensitive to H2O2. Was the denuder efficient in removing gaseous H2O2? How was that determined?
We tested the denuder with two gaseous oxidants, ozone and $NO_2$. We did not observe an increase in signal even at a dose of about 500 ppb over 5 hours. $H_2O_2$ mixing ratios in ambient air are at least two orders of magnitude lower. Therefore, we expect no breakthrough of $H_2O_2$. To be on the safe side we exchanged the denuder for each laboratory experiment (~ 5 h). Further, we checked the ROS blank by measuring the particle-free air (through the particle filter and the denuder) to see if the ROS blank signal increases due to gas phase break through. This was only observed in some wood burning experiments with extremely high pollutant concentrations.
We added in Sect. 3.2 on gas phase interference tests: Based on these results we assume that gaseous $H_2O_2$ is also completely removed.

We also mention in the text how long the denuder was used (page 7): 2) replacing the hydrophilic and hydrophobic filters in the aerosol collector and the denuder every 2-3 days during ambient measurements; while in laboratory experiments, we exchanged the denuder for each laboratory experiment (~ 5 h) to be on the safe side;

p.4, line 12/13. Please support the statement in this sentence with evidence. By how much was the lifespan of the solution shortened and how did the additional contamination affect the measurement?

We tested the auto-oxidation of the working solution containing both HRP and DCFH in the same way as in King and Weber (2013). By mixing only OF-UPW with the HRP-DCFH working solution the signal, which is actually the background, increased with a rate of 0.9%/h. This means that there is a slow reaction with the dissolved oxygen consuming the DCFH and shortening the lifespan of the HRP-DCFH solution.

When the sample is extracted online with the HRP solution as in Fuller et al. (2014), the HRP needs to go through the aerosol collector, where contaminants adsorbed on the hydrophilic/hydrophobic filters or the oxygen in the mist chamber might react with HRP and then oxidize DCFH as described by Berglund et al. (2002) and modified by Fuller et al. (2014). We added this information in the modified manuscript (Page 4): We tested the auto-oxidation of the working solution containing both HRP and DCFH. By mixing only OF-UPW with the HRP-DCFH working solution the signal, which is actually the background, increased with a rate of 0.9%/h. This means that there is a slow reaction with the dissolved oxygen consuming the DCFH consequently shortening the lifespan of the HRP-DCFH solution. When the sample is extracted online with the HRP solution as in Fuller et al. (2014), the HRP needs to go through the aerosol collector, where contaminants adsorbed on the hydrophilic/hydrophobic filters or the oxygen in the mist chamber might react with HRP and then oxidize DCFH as described by Berglund et al. (2002) and modified by Fuller et al. (2014). Therefore, we used only OF-UPW to extract the aerosol samples. The DCFH and HRP reagents were kept separate and were only mixed together right before the aerosol aqueous extract was added.

p. 7, section 2.2 I am not sure this section is necessary as it does not add any information.

We decided to add this section because we have tested many aspects and it might be hard for a reader to quickly grasp all the purpose and the extent of those tests. We think this Section is useful to guide the reader and we would like to keep it.

p.8, line 1: With what experiment was the residence and response time determined.

The residence time was determined from the time of injection of an $H_2O_2$ solution to the time the fluorescence signal started to increase. The response time was determined as the time required for the signal to increase from 10% to 90% of its final value. We added this information in the manuscript on page 8: The residence and response time of the sample in the instrument were measured to be approximately 19 min and 8 min, respectively. The former was determined as the time from injection of an $H_2O_2$ solution to the time the fluorescence signal started to increase while the response time corresponds to the rise time of the fluorescence signal from 10 % to 90 % of the full signal.

Fig. 2: How is the difference in detector response reconciled between the two compounds?

In Fig. 2 we discuss the influence of the reaction time on the detector response of these two compounds $H_2O_2$ and 2-butanone peroxides. Doubling of the reaction time did not increase the signal strength for both compounds. The difference in detector response of various ROS is shown in Sect. 3.1.2 & Fig. 3. Many of the tested ROS have a lower response compared to $H_2O_2$. This has probably to do with the peroxidase, which mediates the reaction of the peroxide with DCFH. However, the mechanism of this is not known and would need detailed investigations, which are outside of the scope of this paper.

At the end of the discussion (page 11) we added the following sentence: Thus, we regard the ROS signal measured by the DCFH assay as a lower limit for the effective ROS content.

p.8, line 8: Was the LOD online determined by using ambient air?
Yes, the online LOD was determined by measuring ambient air passing through the denuder and the particle filter. We clarified this in the modified manuscript (page 9): Under normal instrument operation condition, an instrument limit of detection (LOD) of 2 nmol m$^{-3}$ of sampled ambient air was determined for the online methodology.

p.9 & Fig.S3: Looking at the data in Fig.S3 and confidence intervals shown it look to me the detection limit is more in the range of 15 nM. How does that compare with numbers discussed on p.8.

We assume that you calculated the 15 nM from the confidence interval at very low H$_2$O$_2$ concentrations. We determined the method detection limit by two approaches: 1) injecting several times different batches of OF-UPW (LOD = 9 nM); 2) extraction of different blank filters (LOD = 13 nM). If we calculate an LOD from the calibration curve of Fig. S3 we obtain 18 nM. This LOD would include both the variability of the background (OF_UPW and blank filter) and the conditions of the extraction. We decided to take the largest of these three numbers (18 nM) as the method detection limit. This discussion is added in the manuscript (page 10): Based on this, the uncertainty of H$_2$O$_2$ at extremely low concentrations would be 18 nM. This is larger than the method LOD determined above from the OF-UPW and blank filters. We consider the largest of these uncertainties (i.e.18 nM) as our final method LOD.

p.9, line 20: Did the use of ethyl acetate affect the reactivity of HRP and/or DCFH? How was this verified? The enzyme HRP could be strongly impaired in its reactivity in an organic solvent.

Studies have found that only about 10 % of the initial activity of HRP was lost after incubation in ethyl acetate (EA) for 12 hours. The influence of EA on the HRP activity is much lower than by other organic solvents like acetone, tetrahydrofuran (Jang et al., 2014). To prepare the calibration solutions we first dissolved the pure chemicals in ethyl acetate (EA) and then diluted this solution ~10000 times in water to reach the desired concentration. This means that the EA concentration is roughly 1 mmol/L when mixed with HRP/DCFH. We believe that its influence on HRP/DCFH is negligible at such a low concentration.

p. 10, line 5: Data are only shown from 30-150nM, not 0 – 150nM.
We have added the 0 nM measurements in Fig. 3.

p. 11, Fig. 4: No detail is given for the data shown in Fig. 4. What are the dates, collected? Do they correspond to data shown in the references cited? Why is there a difference in the two data sets, what are the errors on the data shown etc.
Indeed, the description of the sites and filter sampling methods in Milan and San Vittore is given in the references cited. We mention this now (page 12): More details on the analysis of the samples can be found in the cited references. The different slopes between these two data sets might be due to the different emission sources (traffic in Milan and wood combustion in San Vittore) at these two locations as investigated in the cited papers (Perrone et al., 2016; Zotter et al., 2014). We add this information in the modified manuscript (page 12): The different slopes between these two data sets might be due to the different emission sources (traffic in Milan and wood combustion in San Vittore) at these two locations (see Perrone et al., 2016; Zotter et al., 2014).

We also added error bars on the data representing our measurement precision (Fig. 4 & legend).

p.13, line 14: SO4 and NO3 (given in units of ug m-3) were mixed with H2O2 (given in units of nM) for cross sensitivity test. How was that done. Was the sulfate and nitrate nebulised as aerosol? A lot more detail needs to added here.

We prepared solutions of ~ 1.38 µM $SO_4^{2-}$ and ~20 µM $NO_3^-$ from $(NH_4)_2SO_4$ and $NH_4NO_3$, respectively. Such concentrations would typically be received after collection of 23.5 µg m$^{-3}$ $SO_4^{2-}$ and of 228 µg m$^{-3}$ $NO_3^-$ with the online instrument. This is equivalent to 5 and 30 times higher concentrations than the ambient concentration as observed in Milan (Perrone et al., 2016). The measurement of these concentrations is then compared to the mixture of similar concentrations with 115 nM $H_2O_2$ and 272.5 nM 2-BP, as listed in Table 2. To make it more clear, we modified the text and added the prepared concentration of the $SO_4^{2-}$ and $NO_3^-$ solutions (page 14-15): Therefore, we tested the fluorescence response to ~ 1.38 µM $SO_4^{2-}$ and ~20 µM $NO_3^-$ solutions prepared from $(NH_4)_2SO_4$ and $NH_4NO_3$, respectively. Such concentrations would typically be observed after collection of 23.5 µg m$^{-3}$ of $SO_4^{2-}$ and 228 µg m$^{-3}$ of $NO_3^-$ with the online instrument. This is equivalent to ~ 5 and ~ 30 times higher concentrations than observed in Milan (Perrone et al., 2016). These measurements are then compared to cross sensitivity tests of ~1.38 µM $SO_4^{2-}$ and ~20 µM $NO_3^-$ with 115 nM $H_2O_2$ and 272.5 nM 2-BP (Table 2).

p. 14/Fig. 6: The experiments with Fe2+/Fe3+ are discussed in a purely descriptive way. A more detail discussion rationalising the results and referencing Fenton reactions is needed.

We added the reaction equations to the description and give some more explanations. It reads now (page 15):

This might be due to the consumption of a substantial amount of $H_2O_2$ by $Fe^{2+}$, for the production of $HO\cdot$ ($Fe^{2+} + H_2O_2 \rightarrow Fe^{3+} + OH^- + HO\cdot$), which will further react with $H_2O_2$ and resulting in further reduction of the $H_2O_2$ concentration ($HO\cdot + H_2O_2 \rightarrow H_2O + HO_2\cdot$; $HO_2\cdot + H_2O_2 \rightarrow O_2 + H_2O + HO\cdot$)(Kolthoff and Medalia, 1949). This indicates that concentrations of soluble $Fe^{2+} \leq 1$ nM, which were obtained at ambient concentrations of $\leq 10$ ng m$^{-3}$ soluble $Fe^{2+}$ in the online instrument, will not influence the ROS measurement. However, in cases of high ambient soluble $Fe^{2+}$ concentrations the ROS signal might be reduced, whereby this also depends on the $H_2O_2$ equivalent concentration. Measured ambient iron concentrations were found to be in the range of tens to several thousands of ng m$^{-3}$ (Perrone et al., 2016; Oakes et al., 2012; Visser et al., 2015). Oakes et al. (2012) reported that water soluble Fe (II) constitutes between 2.5 and 32 % of total iron, resulting in a water soluble Fe (II) concentration up to 30 ng m$^{-3}$, which would be equivalent to ~ 2 nM in our online instrument. According to our 1$^{st}$ pair of experiments in Fig. 6 (1 nM $Fe^{2+}$ mixed with 113 nM $H_2O_2$ solution) this would not suppress the ROS signal.

p.14, line 11: It is mentioned that water-soluble Fe2+ is measured up to 100s ng m-3 in ambient samples. To what concentration does that correspond in the working solution. Could the ROS signal potentially be suppressed under these conditions? This should be discussed more clearly.
Oakes et al. (2012) report water-soluble $Fe^{2+}$ up to 30 ng/m$^3$, which is equivalent to ~ 2 nM in the online instrument. This concentration is too small to suppress the ROS signal according to our 1$^{st}$ pair of experiments (where we prepared 1 nM $Fe^{2+}$ mixed with 113 nM $H_2O_2$ solution). To further confirm this, we measured the influence of genuine atmospheric particulate metals on the ROS signal in Fig. 7 and the results showed no influence either. We explain this now in the modified version (see answer above).

p. 17/Fig. 8: Are the units for the x and y axis "nmol H2O2 m-3"? If yes, this should be indicated explicitly.
Yes, they are. We modified the units in Figure 8 accordingly.

p.16, line 15/Fig.8: It is mentioned that filters were collected before and after the VACES. Was ROS different in the filters collected before and after the VACES? Did the use of a VACES affect online ROS concentrations?

We did not collect filters before and after VACES at the same time. Thus, a direct comparison cannot be done. We compared the filters collected before and after the VACES with the on-line measurements, which were always done after the VACES. No systematic trend in concentrations was observed. We added this information in the new version of the manuscript (page 18): We did not observe a systematic difference between ROS concentrations on filters taken before and after VACES compared with the online measurements.
During reanalysis of this data, we have noticed that the ROS didn't response to the short changes enrichment factor during the adjustment of the setup, thus we readjusted the calculation of enrichment factors for these periods. This improvement led to minor changes in the concentrations of the online measurement (Fig. 8), but did not affect any of the conclusions.

Fig 9 and related discussion: It is not acceptable to derive a half-life of ROS from the data shown in Fig. 9. The time resolution of the data presented is far too sparse to constrain the half-life of ROS to any reasonable accuracy. This Figure has to be deleted or much more data has to be provided to make a meaningful statement about ROS half-life.

We think the reviewer has raised a valid point. Therefore, in the corrected version of the manuscript we have reconsidered the fitting procedure to carefully account for the uncertainties. The obtained parameters are statistically not different from those obtained before (differences within 10 %) and the related uncertainties and additional limitations of the approach used are now discussed in the corrected version of the manuscript in Sect. 3.4.2 (page 20):

The measured ROS concentrations in SOA from the different wood combustion experiments exhibit a clear decrease with increasing filter storage duration (Fig. 9). In addition, this decay seems to follow a double exponential function. This indicates the presence of a short-lived fraction $A_1$ with a decay constant $\pi_1 = \ln(2)/T_1$ and a long-lived fraction $A_2$ with a decay constant $\pi_2 = \ln(2)/T_2$, where $T_i$ represents the half-life. A biexponential decay function was applied to fit the experimental values, whereby the two decay constants are considered to be the same for all experiments:

$$ROS_{norm}(t) = A_1 * EXP(-\pi_1 * (t - t_1)) + A_2 * EXP(-\pi_2 * (t - t_1)) \qquad \text{Eq. (3)}$$

Here $A_{2,i} = 1 - A_{1,i}$, $0 \leq A_{1,i}$, $A_{2,i} \leq 1$, $i$ refers to an experiment number, $t$ is the time after sampling and $t_1$ is the time when the first offline measurement was performed. $ROS_{norm}(t)$ is the ROS measured at time $t$ normalized to the ROS measured at time $t_1$. The model parameters and their respective uncertainties are shown in Table 3. Measured and modeled values are compared in Fig. 9.

The results show that the two ROS fractions have highly different reactivity. The final modelling yields $\pi_1 = 9.68 \pm 2.56$ and $\pi_2 = 0.0016 \pm 0.0019$. The second fraction (long-lived) appears to be not reactive within our uncertainties and experimental time scales, as the associated reaction rate, $\pi_2$, is not statistically different from 0. The first fraction (short-lived) is highly reactive, with a half-life time $T_1 \approx 1.7 \pm 0.4$ h, similar reaction time-scales and extents were observed for SOA from α-pinene ozonolysis (Krapf et al., 2016). The uncertainty analysis suggests that we are capable of determining the reaction rate of reactive ROS, but not that of the long-lived ROS. The fraction of the long-lived ROS ($A_{2,i}$) could be determined with acceptable errors of 20 %. The main aim of the model is to show that the fraction of unstable ROS may vary significantly between experiments, but could be as high as 75 %, which highlights the need for an online ROS measurement technique. This variability in the

contribution of the unstable ROS fraction could be related to the burning conditions, in this study (shown in Figure S5).

We also discussed the main limitations of the model and this part reads as follows (page 20-21):

The model considers ROS to be composed of two components with different decay rates. However, we do expect that the OA contains spectrum of ROS with a wide range of reactivities. The model is thus a simplification of the ROS in the aerosol. Another simplification is that the decay rates of these two ROS components are considered to be the same between experiments. This may explain the reasons behind the high uncertainties in determining the rates, but does not have a significant effect on the determination of the contributions of the two fractions, $A_{1,i}$ and $A_{2,i}$. We also note that the decay rates and the ROS fractions determined from our results are specific for biomass burning SOA and cannot be extrapolated to other systems.

p. 20, line 4: It should be explained to what "improvements" compares to. Similar instruments by other groups use some of the "improved" conditions as well. Please be precise in your statements.

We added the references of the instruments to which we compared ours (page 23). We also added the minor difference of the second point with the study of Fuller et al. (2014): 2) separation of DCFH and peroxidase working solutions, which were then mixed just before reaction with the sample solution;

p. 20. Line 21: See above. The data presented here cannot support a lifetime estimate.

The reasons we would like to keep it are listed in the previous answer.

References

Berglund, G. I., Carlsson, G. H., Smith, A. T., Szoke, H., Henriksen, A., and Hajdu, J.: The catalytic pathway of horseradish peroxidase at high resolution, Nature, 417, 463-468, 2002.

Charrier, J. G. and Anastasio, C.: On dithiothreitol (DTT) as a measure of oxidative potential for ambient particles: evidence for the importance of soluble transition metals, Atmos. Chem. Phys., 12, 11317-11350, 2012.

Dall'Osto, M., Beddows, D. C. S., Harrison, R. M., and Onat, B.: Fine Iron Aerosols Are Internally Mixed with Nitrate in the Urban European Atmosphere, Environ. Sci. Technol., 50, 4212-4220, 2016.
Fuller, S. J., Wragg, F. P. H., Nutter, J., and Kalberer, M.: Comparison of on-line and off-line methods to quantify reactive oxygen species (ROS) in atmospheric aerosols, Atmos. Environ., 92, 97-103, 2014.

Janssen, N. A. H., Yang, A., Strak, M., Steenhof, M., Hellack, B., Gerlofs-Nijland, M. E., Kuhlbusch, T., Kelly, F., Harrison, R., Brunekreef, B., Hoek, G., and Cassee, F.: Oxidative potential of particulate matter collected at sites with different source characteristics, Sci. of The Total Environ., 472, 572-581, 2014.

Jiang, Y., Cui, C., Huang, Y., Zhang, X., and Gao, J.: Enzyme-based inverse opals: a facile and promising platform for fabrication of biocatalysts, Chem. Commun., 50, 5490-5493, 2014.

Kim, S., Jaques, P. A., Chang, M., Froines, J. R., and Sioutas, C.: Versatile aerosol concentration enrichment system (VACES) for simultaneous in vivo and in vitro evaluation of toxic effects of ultrafine, fine and coarse ambient particles Part I: Development and laboratory characterization, J. Aerosol Sci., 32, 1281-1297, 2001.

Oakes, M., Weber, R. J., Lai, B., Russell, A., and Ingall, E. D.: Characterization of iron speciation in urban and rural single particles using XANES spectroscopy and micro X-ray fluorescence measurements: investigating the relationship between speciation and fractional iron solubility, Atmos. Chem. Phys., 12, 745-756, 2012.

Perrone, M. G., Zhou, J., Malandrino, M., Sangiorgi, G., Rizzi, C., Ferrero, L., Dommen, J., and Bolzacchini, E.: PM chemical composition and oxidative potential of the soluble fraction of particles at two sites in the urban area of Milan, Northern Italy, Atmos. Environ., 128, 104-113, 2016.

Valko, M., Morris, H., and Cronin, M. T. D.: Metals, Toxicity and Oxidative Stress, Curr. Med. chem., 12, 1161-1208, 2005.

Zotter, P., Ciobanu, V. G., Zhang, Y. L., El-Haddad, I., Macchia, M., Daellenbach, K. R., Salazar, G. A., Huang, R. J., Wacker, L., Hueglin, C., Piazzalunga, A., Fermo, P., Schwikowski, M., Baltensperger, U., Szidat, S., and Prévôt, A. S. H.: Radiocarbon analysis of elemental and organic carbon in Switzerland during winter-smog episodes from 2008 to 2012 – Part 1: Source apportionment and spatial variability, Atmos. Chem. Phys., 14, 13551-13570, 2014.